# Nurse Manager Core Competencies: A Proposal in the Spanish Health System

**DOI:** 10.3390/ijerph17093173

**Published:** 2020-05-02

**Authors:** Alberto González García, Arrate Pinto-Carral, Jesús Sanz Villorejo, Pilar Marqués-Sánchez

**Affiliations:** 1Department of Nursing and Physiotherapy, Universidad de León, 24401 Ponferrada, Spain; agong@unileon.es; 2SALBIS Research Group, Department of Nursing and Physiotherapy, Universidad de León, 24001 Ponferrada, Spain; 3Director of the University Dental Clinic, European University of Madrid, 28670 Madrid, Spain; jesus.sanz@universidadeuropea.es; 4SALBIS Research Group, Department of Nursing and Physiotherapy, Universidad de León, 24401 Ponferrada, Spain; pilar.marques@unileon.es

**Keywords:** nurse manager, competence, core competencies, governance, leadership

## Abstract

Nurses who are capable of developing their competencies appropriately in the field of management are considered fundamental to the sustainability and improvement of health outcomes. These core competencies are the critical competencies to be developed in specific areas. There are different core competencies for nurse managers, but none in the Spanish health system. The objective of this research is to identify the core competencies needed for nurse managers in the Spanish health system. The research was carried out using the Delphi method to reach a consensus on the core competencies and a Principal Component Analysis (PCA) to determine construct validity, reducing the dimensionality of a dataset by finding the causes of variability in the set and organizing them by importance. A panel of 50 experts in management and healthcare engaged in a four-round Delphi study with Likert scored surveys. We identified eight core competencies from an initial list of 51: decision making, relationship management, communication skills, listening, Leadership, conflict management, ethical principles, collaboration and team management skills. PCA indicated the structural validity of the core competencies by saturation into three components (α Cronbach >0.613): communication, leadership and decision making. The research shows that eight competencies must be developed by the nursing managers in the Spanish health system. Nurse managers can use these core competencies as criteria to develop and plan their professional career. These core competencies can serve as a guideline for the design of nurse managers’ development programs in Spain.

## 1. Introduction

Economic and social changes have led to an adaptation of healthcare management at all levels and a change in the way in which services are provided [1,2,3,4,5,6]. The relationship between the economy and sustainability should be causes that make it necessary to develop management competencies with a high level of development, because these competencies are related to higher performance and outcomes [7,8,9,10]. To address these changes, nurses need to be part of the core of healthcare [2,4,11,12]. This claim is justified because the nurse is a professional with a high degree of leadership in many healthcare processes [13], because of their closeness to patients, families and the community [14]. Multiple research programs are led by nurses, including oncology, mental health, patient safety, palliative care and childcare, among others, which shows the importance of the nurse in the healthcare system [15,16,17,18,19]. For this reason, the participation of the nurse in the governance of healthcare organizations is recognized as fundamental, both for health outcomes and the sustainability of the health system [20,21,22,23].

The American Institute of Medicine, in its report entitled “The future of nursing: leading change in health”, already identified nursing in 2010 as a key player in critical decision making and in the transformation of healthcare [24]. In the same way, Weber et al. [25] and McClaringan et al. [20] express themselves by stating that nursing is fundamental to the implementation of shared governance, because their commitment and participation is essential for the sustainability and improvement of health. Panayotou et al. [26] emphasizes nursing within strategic plans, so that their key actions are focused on the creation of a culture of good practice. Brooks-Cleator et al. [27], in his studies on transculturality and governance, focus on the importance of the nurse in establishing a culture of safety. The involvement of nurses in social action committees has achieved a great impact on the everyday problems of people [28], highlighting the importance of their participation in decision making [29]. When the nurse is involved in the different parts of the healthcare process (management and nursing care), better results are achieved [30], communication is enhanced, together with collaboration between the different professional groups, innovation, organizational commitment and retention of staff [23].

Nurse leaders and nurse manager are different roles, although there is a natural overlap of the required competencies [1,10,31]. To develop the role of the nurse in the governance of healthcare organizations, the Global Nursing Leadership Institute (GNLI) in the actions of Nursing Now (2020) has developed a nurse preparation program designed to promote leaders for change, focused on new policies to improve the health of the population [18,32]. Thus, to identify, mentor and train nurse leaders, the management competencies are an essential resource [31]. Furthermore, for the integration of nurses at different levels of organizational management and governance, the nurse should develop management competencies that go beyond the scope of nursing [33,34,35]. 

The nurse manager is responsible for planning and managing resources, organizing nursing care, supporting teamwork, evaluating the services provided, and contributing to the achievement of optimal results for the both the organization and the patients [36,37]. Based on the literature review, it is necessary to increase the knowledge of the role of the nurse manager [36,38,39,40,41], because the necessary competencies are often not clearly defined [35,42], which would explain this lack of conceptualization of the nurse manager’s role. This same absence is evident in the Spanish context since there are no core competencies for carrying out management functions.

Core competencies are the collective learning of the organization, especially with regard to skills related to the generation of a product or service, so that all necessary knowledge and technologies are integrated [43,44]. The core competencies in nursing management are associated with the success of the healthcare organization [45]. Therefore, the core competencies for nurse managers are the set of fundamental competencies needed to ensure their work effectiveness [46]. There are three functional roles within nursing management, the operational nurse manager (performs his or her function at unit level), the logistic nurse manager (performs his or her function at the department level), and the top nurse manager (performs his or her function at the organizational level); all of them should develop the core competencies to help improve the quality of healthcare. [46].

Hence, the main objective of this study is to propose the core competencies to be developed by the nurse manager in the Spanish health system. To achieve this objective, the following specific objectives were set: To determine core competencies for each functional level of nursing management by expert consensus;To describe the level of development of competencies for each functional level of nursing management by expert consensus;To describe the training needed to develop each of the required competencies by expert consensus;To evaluate the structural validity of the proposed core competencies.

This paper is laid out as follows: first, the current state of knowledge about the nurse manager is described. The second part explains the research methods. Next, the results are presented and interpreted. Finally, the paper includes a discussion and conclusion.

## 2. Materials and Methods

### 2.1. Review Literature

Based on a scoping literature review during 2018-19 to identify existing competencies related to nurse managers, electronic databases were used (Web of Science, Scopus, PubMed and CINAHL) to conduct the search, identifying 56 competencies for nurse managers. Relevant studies were identified, such as that carried out by the American Organization of Nurse Executives (AONE) who established two competency models for nurse managers [47]. In addition, the literature review identified other important research to define competencies—for example, the Chase instrument [48] or the research carried out by Kantanen [42], DeOnna [35] or Pillay [49], among others. The results of this review are the foundation for the execution of the current Delphi study, which assesses the competencies for nurse manager positions.

### 2.2. Delphi Methodology

The study was carried out through four rounds of the Delphi method. The Delphi method is a method used to obtain a consensus from a group of experts [50], where the overall view will provide more solid information than that offered by a single person on an individual basis, thus reducing the subjectivity [50,51]. The questionnaires were administered through the LimeSurvey online platform. The questionnaires included an instruction form for the expert, the authorization to participate in the research and the instructions. A reminder email was sent every 4 days until a reply was received. After the survey, the researchers selected or excluded the items that received less than 80% agreement among experts.

The objective of the first Delphi round was to reach a consensus among the panel of experts about the core competencies for nurse managers. During the second round, the experts were asked individually if they wished to reconsider their opinions in light of the feedback. In the third round, the experts were asked to provide a consensus about the core competencies at each nurse manager on a functional level. The experts also agreed on the training required for each level of competence (expert, very competent, competent, novice advanced and novice). The fourth round allowed experts to reconsider their opinions in view of the feedback from the third round. 

#### 2.2.1. Consensus

In any Delphi study, the definition of consensus should be set a priori. Thus, for this research, we defined the consensus in three ways: (I) if at least 80% of the experts agreed with the competencies, responding “agree” or “complete agreement” in the questionnaires; (II) if at least 80% of the experts agreed with the degree of development of the competencies; (III) if at least 80% of the experts agreed with the type of training required. Where an agreement was not reached, items were deleted for the next Delphi round.

#### 2.2.2. Participants

In this study, we decided to invite experts from two categories and twelve groups: experts in health management (Table 1) and experts in the health environment (Table 1), because experts in these two categories have valuable knowledge on nursing management.

#### 2.2.3. Variables

The variables of the study were:Socio-demographic variables: To define the profile of the experts, information was collected related to age, sex, profession, university education, postgraduate education, professional role, place of study, years of professional practice, years of management experience, management functions performed and international experience;Competencies: From the review of the literature emerged the list of competencies to be proposed for the experts.

#### 2.2.4. The Delphi Questionnaires

Two questionnaires were developed ad hoc as measuring instruments. 

Competencies needed for nurse managers (Appendix A): Each participant rated his/her level of agreement or disagreement with each competency according to a one to five Likert scale (1 = complete disagreement, 5 = complete agreement);Level of competency development for nurse managers (Appendix B): In order to reach a consensus about the level of development of the competencies at each level of management, the degree of agreement or disagreement with each competency according to a one to five Likert scale (1 = novice, 5 = expert), and the type of training required to develop the competencies, according to a one to six Likert scale (1 = University Extension Diploma, 2 = Continuing education, 3 = University Expert, 4 = University specialization diploma, 5 = master’s degree, 6 = Ph.D.) was recorded.

#### 2.2.5. Level of Development

For this research, the term “level of development” was used to refer to the level of deepening in each competency that the nurse manager should acquire in each of the functional levels, thus the level of development would be:Novice: follows the rules and plans;Novice advanced: can provide partial solutions to unfamiliar or complex situations;Competent: strong demonstration of competency;Very competent: significant demonstration of competency;Expert: when demonstrating the behavior of the competency model.

#### 2.2.6. Validity and Reliability

The validity and reliability of the questionnaires was carried out with a group of 12 people selected on the basis of the same criteria used for the panel of experts. The reliability of the questionnaires was ensured by carrying out a Cronbach’s Alpha Coefficient analysis. The content validity was estimated through expert judgement, which analyzed errors and ambiguities in the formulation of the questions, excess items, proposals for improvement, suggestions for the style of the surveys. 

### 2.3. Principal Component Analysis

The Principal Component Analysis (PCA) is a data transformation technique. The aim of the method is to reduce the dimensionality of multivariate data, while preserving as much of the relevant information as possible [52]. The factor analyses were carried out with respect to the theory of Thurstone [53,54] (3 phases): first, the assessment of the adequacy of the data for factorial analysis, second, the extraction of factors, and finally the rotation and interpretation of factors.

For determining the suitability of the data for factorial analysis, we used the Kaiser–Meyer–Olkin (KMO) test. The next step was the extraction of factors, using Kaiser’s criteria, which makes the decision based on an eigenvalue greater than one [55], and a scree plot, which is a graphical representation of the eigenvalues. This graph helps to find the inflexion point and the number of factors above this point that should be retained [56]. Finally, we proceeded with the rotation and interpretation of the factors, through the varimax rotation method and Kaiser standardization, to achieve a structure as simple as possible that was easy to interpret [57].

## 3. Results

### 3.1. Demographics of the Expert Panel

A total of 50 experts consented to participate and took part in the Delphi study. Table 1 lists the demographic characteristics of the complete expert panel. The response rate for all of the Delphi rounds was 100%.

### 3.2. Delphi Study

During the first and second Delphi rounds, 51 competencies were agreed by consensus (more than 80%) from the proposed list. In round 1, the percentage of “total agreement” was 100% (“agreed” or “complete agreement”) with the competencies decision making, communication skills, listening and conflict management. In this round, more than 80% of the experts were in “total agreement” with eight competencies (Table·2): decision making, communication skills, listening, leadership, conflict management, ethical principles, collaboration and team management skills.

Experts in round 2 were provided with individual feedback from the round 1 survey. This feedback included the complete expert panel responses. Participants were asked if they agreed or disagreed with the statements that were made in the previous round. From round 2, it appears that the experts showed a “complete agreement” with a percentage equal to 100% in the competencies identified as the core competencies (Table 2). In round 2, the competencies with less than an 80% consensus were eliminated.

During the third and fourth Delphi rounds, the eight competencies from the core competencies were shown to be necessary for the three levels of nurse manager existing in Spain (operations, logistics and top management), differing in the level of development of the competencies at each level of management (“Expert”, “very competent” and “competent”). The panel of experts in round 3 were asked about the level of development of each competency to reach a consensus. The experts in round 4 were again provided with individual feedback from round 3, and asked to indicate their agreement with statements that were made by participants in the previous round. The final consensus is shown in Table 3.

During the third round, the experts were asked to indicate their opinion about the appropriate training to reach the right level of competency. In round 4, experts were again provided with individualized feedback from the previous round. The final consensus is shown in Table 4.

### 3.3. Principal Component Analysis

The data were suitable for factoring as the correlation matrix showed a predominance of meaningful results (*p* < 0.05), Bartlett’s test was significant (*p* < 0.001) and KMO value was 0.505. The integration of competency listening into the competency communication skills was appropriate for factoring (as shown in Table 5).

The extraction of factors showed three factors that explained 68.67% of the total accumulated variance. The varimax rotation method yielded a three-factor solution: communication, leadership and decision making (Table 5). The observed convergence between the Kaiser criteria and the scree plot adds certainty to the results. The reliability of the core competencies showed a Cronbach’s alpha value of 0.613, indicating a satisfactory result. (Table 5).

## 4. Discussion

This paper reports on the findings of the core competencies to be developed by the nurse manager in the Spanish health system. Decision making, relationship management, communication skills, listening, leadership, conflict management, ethical principles, collaboration and team management skills were seen as the core competencies for nurse managers. These findings are consistent with the findings from previous studies [47,51,52]. Kantanen et al. [42] emphasizes competency in decision making as a critical competency. McCarthy [58] highlights core competencies that are aligned with our research in communication, relationship management, ethical values and decision making. Our research is also aligned with Pillay [59] and with Gunawan [60], when he described relationship management, conflict management and collaboration and team management as basic competencies.

To emphasize the strength of the core competencies identified in this research, we should say that this were also identified through a scoping review of the literature, with the exception of the listening skills and ethical principles, which were not found among the most frequent results in the review. It should be noted that the frequency of citation in the selected articles was used as a criterion for identifying the core competencies in the literature review.

This study identified that all the core competencies are needed independent of the functional level of nurse managers (executive management, logistics and operational management). Although each functional level requires different levels of competency development to be reached. There was a consensus between the experts in Delphi rounds 3 and 4. These findings are consistent with findings from previous studies (e.g., [47,58]). McCarthy et al. [58] highlighted that core competencies should be common to all three levels of management at different degrees of development. The AONE [61] has shown shared competencies in their different models (“Nurse Manager Competency”, “Nurse Executive Competencies” and “Nurse Executive Competencies: CNE system”), and in the Nurse Executive Competency Assessment Tool, which differentiates the degree of development of each competency. In another sense, the AONE also defined non-shared competencies in their models.

With regard to the development of competencies, an agreement was reached in Delphi rounds 3 and 4. The experts agreed that competencies should be developed at “competent” (this was considered to have been reached when there is a strong demonstration of competency), “very competent” (level reached when there is a significant demonstration of competency) and “expert” level (level reached when it demonstrates the behavior of the competency model). This proposal is in agreement with AONE, who use the levels competent, proficient and expert for the development of competencies, emphasizing how these levels are reached through master’s degree studies or a Ph.D. [62,63]. In contrast, in other studies such as “Nurse manager competencies”, the focus is on the degree to which the competencies contribute to the nurse manager’s work (minimally, moderately, significantly and essentially) [48]. Furthermore, the results of the current study emphasize the need for a high level of competence development, in the same way that Crawford et al. [64] demonstrated by indicating how executive practice would require a high degree of specialization and a specific development of competencies.

During Delphi rounds 3 and 4, the expert panel achieved a consensus about the training to be developed by the nurse manager on the three competency levels (“expert”, “very competent” and “competent”). The “competent” level is reached through continuing education, University Expert and University specialization diploma. With regard to the “very competent” level, the consensus was reached with University expert, University specialization diploma or master’s degree. Finally, the “expert” level is reached through master’s and Ph.D. studies. We should keep in mind how work experience and education significantly influence the development of competencies of nurse managers [65]. However, experience as a nurse manager does not prepare them for the wide range of skills needed, requiring specialized training and work experience in concrete situations [33,66]. Learning experientially as a nurse manager should be accompanied by prior planning and close mentoring [67]. Previous studies show that the quality and level of training are responsible for orienting nurse managers towards good governance and the acquisition of the global vision of the organization [33,68,69]. Furthermore, the results of the current study emphasize how it is possible to appreciate differences between nurse managers who have completed advanced management programs with respect to others who have not participated in this type of training program, adding evidence to previous studies [68,70]. We share the recommendations given by the Joint Commission for Accreditation of Healthcare Organizations regarding the development of different career levels for nurses according to their level of education, training and experience [71]. In addition, and just like Fralic [72] affirmed, our results suggest that the training received by nurse managers would be one of the key aspects, because they are responsible for managing the area with the largest number of people, to make decisions about resource management and other areas such as quality of care, patient safety, research, training, expenditure or investment. The present study, along with other previous studies such as the research carried out by Herrin et al. [73], support that the master’s degree training allows the nurse manager to be able to carry out adequate decision making, as well as for the effective management of health processes. In the same way, Rizani et al. [74] point out that the average competence of nurses is higher when they have developed advanced studies (master’s degree or Ph.D.), increasing with time their level of competency to a higher degree than those nurses who have not developed advanced training.

The PCA verified the core competencies by defining three principal components named communication (communication skills, relationship management, conflict management), leadership (leadership and team management skills) and decision making (decision making and ethical principles), which would therefore constitute the competency factors to develop the role of a nurse manager in Spain (Table 6). The strength of the eigenvalue confirms the importance of the relationship between decision making and ethical principles [75], the need for strong leadership in working groups [76] and communication as a fundamental element in conflict resolution [77]. By comparing the core competencies emerging from our research with the most relevant international studies into core competencies for nurse managers, we would find a shared factor with communication, which should indeed be presented as a shared factor [25,78,79]. 

The communication ability that would be expected from the nurse manager should include the ability to convey critical thinking and generate reflection in nurse teams prior to action [36]. In the same way, it should, for example, facilitate conflict resolution and shared decision making, as well as creation, participation and team management [80]. The differences that would arise between all the core competencies could be related to the different health contexts in which management practice takes place [58].

The purpose of this study was to determine the core competencies for each functional level of nursing management by expert consensus. Our research contributes strong and important evidence to the nursing management field. Firstly, we provide a baseline of competencies for nurses who intend to carry out functions as nurse managers. Secondly, our study is also useful as a tool for evaluating and detecting areas for improvement for nurse managers. Finally, the core competencies should be useful for planning the professional development of nurse managers.

With regard to the limitations of this research, we should mention the different healthcare contexts from which the body of research knowledge is derived. Nursing management has specific characteristics for each of these contexts.

## 5. Conclusions

This study found core competencies for nurse managers in Spain. The successful nurse manager should develop all these competencies (as relevant to their practice) in today’s rapidly evolving healthcare system. In conclusion, this study yielded a consensus on eight core competencies for nurse managers in Spain: decision making, relationship management, communication skills, listening, leadership, conflict management, ethical principles, collaboration and team management skills, oriented towards leadership and good governance of health organizations, and on the basis of the social responsibility of health professionals. The nurse manager is responsible for the largest area of a healthcare organization, managing large budgets and large numbers of nurses. Therefore, a nurse should not be promoted to the role of a nurse manager without advanced management training.

Our research shows the precise level of development of each competency for the different functional levels of nurse manager. The nurse manager at any functional level should develop these core competencies before being promoted to other roles as a nurse manager.

Any nurse who wishes to develop his or her professional career as a nurse manager should first develop the core competencies shown here.

Moreover, our research shows the necessary education required to acquire the competency development necessary for each different nursing management role. Both nurses who want to be promoted to nurse managers and current nurse managers should follow the educational programs shown in order to adapt their knowledge to the requirements of the role.

These core competencies may have implications for practice, organizational policy, and education related to nursing management. The proposed core competencies may contribute to nurse manager role design, selection processes, and nurse manager curriculum design for traditional academic institutions and organizational continued professional development programs. Further understanding of core competencies is likely to inform interventions, which may improve nurses’ work environment, patient care, patient safety and organizational outcomes.

The following research should develop the characteristics corresponding to each of these competencies and training situations.

## Figures and Tables

**Table 1 ijerph-17-03173-t001:** Socio-demographic data from the panel of experts.

Demographic Variable			Frequency
**Gender**			
		Female	32 (64%)
		Male	18 (36%)
**Age**			
		Mean	49.52
		Standard deviation	11.02
		<40 years	10 (20%)
		41–50 years	15 (30%)
		51–60 years	18 (36%)
		>60 years	7 (14%)
**Education**			
		Master’s degree	34 (68%)
		Ph.D.	14 (28%)
**Scope of representation**			
**Expert group**	Group 1	Minister of Health	3 (6.1%)
	Group 2	Head of the Health Department	5 (10%)
	Group 3	General Council of Nurses	3 (6%)
	Group 4	Scientific Association	4 (8%)
	Group 5	Trade Union	3 (6%)
	Group 6	General Manager	5 (10%)
	Group 6	Medical Director	2 (4%)
	Group 6	Nurse Executive	5 (10%)
	Group 6	Management Director	1 (2%)
	Group 7	Middle Nurse manager	2 (4.1%)
	Group 8	Nursing supervisor	3 (6.1%)
	Group 9	Nurse	3 (6.1%)
	Group 9	Doctor	2 (4.1%)
	Group 9	Assistant Nursing Care Technician	2 (4.1%)
	Group 10	Nursing Degree Students	2 (4.1%)
	Group 11	Research/Teaching	4 (8.2%)
	Group 12	Lawyer	1 (2%)

Source: own elaboration.

**Table 2 ijerph-17-03173-t002:** Core of competencies.

Competency	Total Agreement	
	Round 1	Round 2
Decision making	100%	100%
Relationship management	84%	100%
Communication skills	100%	100%
Listening	100%	100%
Leadership	84%	100%
Conflict Management	100%	100%
Ethical principles	80%	100%
Collaboration and team management skills	88%	100%

Source: own elaboration.

**Table 3 ijerph-17-03173-t003:** Development of core competencies at each level of nurse manager.

Competency	Top Management	Logistics	Operations
Decision making	Expert	Very competent	Very competent
Relationship management	Very competent	Very competent	Very competent
Communication skills	Expert	Very competent	Expert
Listening	Expert	Very competent	Expert
Leadership	Expert	Competent	Expert
Conflict Management	Very competent	Very competent	Very competent
Ethical principles	Expert	Very competent	Expert
Collaboration and team management skills	Expert	Very competent	Expert

Source: own elaboration.

**Table 4 ijerph-17-03173-t004:** Training required by competency level.

Competency Level	Type of Training (Consensus)
Novice	University Extension Diploma (100%)
	Continuing education (98%)
Novice advanced	University Extension Diploma (90%)
	Continuing education (90%)
	University Expert (90%)
Competent	Continuing education (96%)
	University Expert (100%)
	University Specialization Diploma (96%)
Very competent	University Expert (96%)
	University specialization diploma (96%)
	Master’s degree (96%)
Expert	Master’s degree (96%)
	Ph.D. (96%)

Source: own elaboration.

**Table 5 ijerph-17-03173-t005:** Principal Component Analysis (PCA) of core competencies.

	CP 1	CP2	CP3	
Communication skills	0.851			
Relationship management	0.771			
Conflict Management	0.620			
Leadership		0.877		
Collaboration and team management skills		0.841		
Ethical principles			0.773	
Decision making			0.706	
Explained variance	28.43%	22.665%	17.576%	
Eigenvalue	1.99	1.587	1.230	
α Cronbach				0.613

Source: own elaboration. Caption: communication (CP), leadership (CP2), decision making (CP3).

**Table 6 ijerph-17-03173-t006:** Comparison of core competencies.

Core Competencies	Leach et al. [79]	Weber [25]	Aone [47]
Communication	Organizational Management	Influence	Leadership
Leadership	Interpersonal effectiveness	Emotional Intelligence	Professionalism
Decision making	Systemic thinking	Result orientation	Communication
	Creative thinking	Change Management	Leadership
	Technical skills	Communication	Knowledge of the health environment
	Ability to adapt	Management vision	
	Customer Service		
	Personal domain		

Source: own elaboration.

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
