# Peer review of "Nurse Manager Core Competencies: A Proposal in the Spanish Health System"

_ijerph, 2020, doi:10.3390/ijerph17093173_

Round 1
Reviewer 1 Report
Dear Authors.
you made an interesting paper.
Here are some ideas for improvement.
Apologies for being late, these times are difficult.
So here it goes:
1 - Introduction - end it with a sentence describing the paper. So, something like - In order to answer the research questions the paper will be composed by the following section - Literature review, Methods, Results, Discussion, Conclusions.
2 - Literature review - this section is missing. What is known about the research questions ? You have something of this in the Discussion in Table 6. You should move and develop. A strong literature review is important in any paper.
3 - Methodology - The Questionnaires you mention should come in annexe of the paper. I got a bit lost without knowing what you asked,
4 - Resuts - ok well done.
5 - Discussion - reflect on the relation of your results with the literature review you will present in the second section. Also indicate implication and limitation. Your three factors are very strong but is this a surprise that Decision making requires ethics, Leadership is related with collaboration and team management skills and communication links to conflict management and relationship management ? What is the value added of your research ?
6 - Conclusion - please summarize results in short paragraph. Please indicate ideas to further research,
Good luck
Reviewer 2 Report
The approach of identifying core competencies though statistical treatment of expert interviews with the Delphi method is interesting. However, the article would improve its impact by addressing two issues:
- First, most competency based development programs are structured along, typically: Leadership/people/behavioral skills and technical skills, the so-called soft and hard skills. This article jumps directly into the subset of soft skills. It would be better to clarify this reduction. Normally, a competence matrix is built on the job description. In the case of nurses, this is certainly true as well. This part should be expanded and clarified in the introduction. E.g. line 117, how did you select the competencies that were proposed to the experts panel. Also; beware of the wording core-competency (see later in the text), that typically means something different in business jargon.
- Second, probably more important, you equate skill maturity (competencies level) with university degrees. How and why do you acquire a expert level in e.g. conflict management with a PhD in e.g. biochemistry? If this would be so, it would be probably known…. If you want to maintain this view, you have to elaborate and make it credible.
Now, in terms of conclusions, this study is still a bit banging open doors, but the methodology provides some interesting avenues.
More specifically:
To: Abstract:
L16: define what you mean with core competencies: does it equate with soft skills (or the like), then reword it less ambiguously. The traditional meaning of core competencies is rather related to the competencies required from a company as competitive differentiation. You mean something else, so define it.
L18: define construct validity with PCA: clarify what you mean or what you did
To 1: Introduction
L31+ff: good introduction to the importance of the nurse profile and requirements. But you do not address a fundamental element of the definition of the required competencies, this is the success (and failure) factors of the job (Job success factor, in the HR jargon) , which normally relates to the job description.
L58: here, you define the so-called management competencies, but you seem to equate this to leadership competencies. Normally, this is understood differently.
L67-72: there, you come to the definition of the requirements of the ‘fundamental’ competencies, but again, you have a rather generic approach, not related to a job profile.
To 2: Materials and methods
L160: in round 1, …100% agreed…: why do we read different figures in table 2: may be clarify possible misunderstanding
L171: In the list of competencies, you do not mention stress resilience, something one would intuitively expect from nurses in operations, any reason why? The exact point is: how do you demonstrate that your skill set is sufficient and comprehensive as success factor for managers-nurses?
L187: As mentioned in the summary above, the logic of this part is difficult to follow. You need to justify why and how leadership can be acquired through a master or PhD in nursery or science, that actually train very different competencies.
L189+ff: OK with the PCA, but is this helping to define a development and training plan?
L199: Table 5: you mention 8 competencies in the text, but list here only 7. For the credibility of your text, correct.
To 4: Discussion
L206: findings from previous studies: references needed, or drop the statement. The following part of the paragraph refers to the definition of competencies by other associations. It would be adequate to carry this discussion over, and have a section on the definition of the competencies, soft and hard skills, mental skills, technical skills, as viewed from different organizations, and then discuss and justify a screening from this landscape of competencies to finally come to your model of 8. Here, you put the cart before the horses.
L230+ff: if you intend to make out of this paper a guide for competencies development for nurses, this para should be revisited and expanded. A maturity level is the rating of an ability to perform or act in a given way, etc… , not a university degree. You would gain in credibility to better elaborate on these points, and also better elaborate on training. Leadership is mostly trained through deliberate practice with coaching and feedback (inclusive 360⁰) , but how you achieve this through university training is, to say the least, not clear. And there are specific leadership and communication training that are probably more appropriate.
But if you want to maintain this view, you have to make it credible in your text.
L265: keep in mind that ‘communication’ and ‘leadership’, without further definition, are ‘mish-mash’ concept that embrace a bit if everything, and are not sufficient for defining a training program. Therefore, no wonder they emerge from a PCA….
To 5: Conclusion
Your conclusion is rather shallow. It could be revisited based on the comments above, and also provide a clearer message on what should be changed in the present education and people development system.
Reviewer 3 Report
Summary and recommendation
A well written paper with a sound methodology. With at few minor revisions the paper is ready to publish.
Major issues
I have no major issues with the paper.
Minor issues
There are some minor issues to attend to in the paper:
Line 78: I would like more explanation of “degree of development”. Perhaps a better wording would be to use "level of development" which you use in the discussion (e.g. line 234 & 235). It is hard to understand the exact meaning of “degree of development”. Perhaps use some of the text in the discussion section (line 230-237) earlier in the paper to help the reader understand what you mean.
Line 125: see above (“degree of development”)
Line 175: see above (“degree of development”)
Line: 222: says These: change to There, “There were consensus…”
Line 295: says This: change to These, “These core competencies…”
Check your references one more time. Not complete information on all entries.
Round 2
Reviewer 2 Report
Dear authors,
The text revisions provide many substantial improvements to the article, that will certainly improve its impact.
The conclusion has now more substance, and can provide hints on how to use this article.
In retrospect, you might even gain on impact that your article is addressing the incremental competencies that a nurse manager should develop and acquire in order to fulfil successfully her/his new position.
This is clear from the conclusion, and you may wish to bring a similar message at the introduction/abstract as well.
some minor points:
L.34-37: try to polish the sentence
L.77-80: Look if talking of incremental compentencies would clarify your message (see above)
L.154: remove once "was used"
L.214: you have very high requirements, because expert level means normally you train on the subject, and contribute to shaping this competence (articles, communication, etc...)
But overall, the article should have a good impact and be helpful to the community of nurses.
